# Towards Theoretical Understanding of
# Sequential Decision Making with Preference Feedback

**Simone Drago** [1]  **Marco Mussi** [1]  **Alberto Maria Metelli** [1]

## Abstract

The success of sequential decision-making approaches, such as *reinforcement learning* (RL), is closely tied to the availability of a reward feedback. However, designing a reward function that encodes the desired objective is a challenging task. In this work, we address a more realistic scenario: sequential decision making with preference feedback provided, for instance, by a human expert. We aim to build a theoretical basis linking *preferences*, (non-Markovian) *utilities*, and (Markovian) *rewards*, and we study the connections between them. First, we model preference feedback using a partial (pre)order over trajectories, enabling the presence of incomparabilities that are common when preferences are provided by humans but are surprisingly overlooked in existing works. Second, to provide a theoretical justification for a common practice, we investigate how a preference relation can be approximated by a multi-objective utility. We introduce a notion of preference-utility compatibility and analyze the computational complexity of this transformation, showing that constructing the minimum-dimensional utility is NP-hard. Third, we propose a novel concept of preference-based policy dominance that does not rely on utilities or rewards and discuss the computational complexity of assessing it. Fourth, we develop a computationally efficient algorithm to approximate a utility using (Markovian) rewards and quantify the error in terms of the suboptimality of the optimal policy induced by the approximating reward. This work aims to lay the foundation for a principled approach to sequential decision making from preference feedback, with promising potential applications in RL from human feedback.

[1]Politecnico di Milano, Milan, Italy.
Correspondence to: Simone Drago <simone.drago@polimi.it>.

*Proceedings of the 42$^{nd}$ International Conference on Machine Learning*, Vancouver, Canada. PMLR 267, 2025. Copyright 2025 by the author(s).

## 1. Introduction

In the last decade, *reinforcement learning* (RL, Sutton & Barto, 2018) has demonstrated great success tackling sequential decision-making under uncertainty with notable results in industrial plant control (Nian et al., 2020), robotics (Kober et al., 2013), clinical trials (Coronato et al., 2020), autonomous driving (Kiran et al., 2021), videogames (Mnih et al., 2015), and, more recently, language models (Du et al., 2023). In RL, the learning process is guided by a *numerical* feedback (i.e., a *reward* function). The reward is often defined informally as "the most succinct description of a task" (Ng & Russell, 2000). More formally, the power of a reward function is apparent since it allows, under the Markovian property of the environment (Puterman, 2014), to approach the learning problem with desirable computational (Papadimitriou & Tsitsiklis, 1987; Littman, 1995) and statistical (Azar et al., 2012) properties.

Nevertheless, the limits of learning with a reward are well known. In the common practice, the reward function is typically designed by a system expert who leverages their domain knowledge to capture the intuitive notion of "solving the task". However, in many real-world scenarios, crafting a reward function that appropriately encodes the desired *objective* can be challenging. Indeed, rewards should go beyond merely capturing the desired *behavior* to enhance their generalizability, interpretability, and transferability to new environments (Ng & Russell, 2000). Defining a reward, often referred to as *reward engineering* (Dewey, 2014), is typically a trial-and-error process involving successive refinements since the behavior learned by the agent can be highly sensitive to misspecifications of the reward (Pan et al., 2022). As such, the choice of the reward function has a critical impact on the success of the agent in learning how to solve the task. Even accepting the availability of a reward function, the community has recently questioned *whether* a reward function is truly an appropriate *mathematical tool* to encode the notion of a goal. The debate dates back twenty years, when Sutton postulated that "all of what we mean by goals and purposes can be well thought of as maximization of the expected value of the cumulative sum of a received scalar signal (reward)" (Sutton, 2004). More recently, this hypothesis has been under investigation, although a defini-

tive answer is currently lacking (Silver et al., 2021; Glukhov, 2022; Vamplew et al., 2023; Bowling et al., 2023).

Why not get rid of the reward? One solution is to ask a human expert for *feedback* on the agent's behavior rather than requiring them to define a numerical reward function. The agent can then learn a behavior that aligns with the expert's *preferences*. In the literature, this paradigm is known as *preference-based reinforcement learning* (PbRL, Fürnkranz et al., 2012). Although PbRL dates back more than twenty years, it has received renewed attention from the community thanks to the rise of *large language models* (LLMs, Zhao et al., 2023a). Indeed, modern LLMs are (pre-)trained using large amounts of data collected by eliciting pairwise human preferences (Ramachandran et al., 2017; Radford, 2018). An established approach for leveraging human preferences is *reinforcement learning from human feedback* (RLHF, Christiano et al., 2017; Stiennon et al., 2020; Bai et al., 2022; Ouyang et al., 2022), which consists of two steps: first, preferences over trajectories are used to learn a reward model, and then, RL is applied using the recovered reward function. In addition to its remarkable empirical performance, RLHF has recently gained a theoretical understanding (Xu et al., 2020; Chen et al., 2022; Saha et al., 2023; Zhan et al., 2024a;b). Nevertheless, these works are closely tied to the assumption of the *existence of an underlying (hidden) numerical signal* (either a proper reward function or a utility defined over trajectories), of which the preferences expressed by the human are an indirect stochastic manifestation.[1] More in general, estimating a *scalar* numerical signal, like in RLHF, from preferences hinders the complexity of the human feedback such as the possible *multi-objective* nature of the human behavior (Hayes et al., 2022). Other approaches focus on learning the policy directly from preferences without going through a reward model (An et al., 2023; Zhao et al., 2023b; Rafailov et al., 2024; Azar et al., 2024). Despite the promising results, these approaches, similar to RLHF, are based on a probabilistic model of human preferences that the learned policy tries to replicate.

Despite the wide variety of approaches, to the best of the authors' knowledge, there is still limited theoretical understanding of the challenges and opportunities involved in learning from preference feedback. In the PbRL literature (Wirth et al., 2017), an agent can roughly operate in three ways: ($i$) learn the policy directly from preferences, ($ii$) estimate a surrogate utility (i.e., a non-Markovian reward) defined over trajectories, or ($iii$) derive a (Markovian) reward function. Moving from ($i$) to ($iii$), we trade off representational power with tractability. On the one hand, ($i$) constitutes a more general approach where no numerical signal needs to be modeled, and as such, could inherently

represent incomparabilities (i.e., situations where the human expert is unable to compare certain pairs of trajectories). However, the definition of optimality, as we will discuss later in the paper, may pose important computational limitations. On the other hand, ($ii$) and ($iii$) are based on a numerical signal and, for this reason, introduce a bias[2] and the need for *multi-objective* signals (Hayes et al., 2022) to model incomparabilities. The positive counterpart of using a numerical signal is that optimality notions (e.g., Pareto optimality, Censor 1977) are well-defined. Nevertheless, planning with general utilities ($ii$) is still intractable, whereas when using rewards ($iii$) coupled with the Markov property, the computation of the optimal policy can be done efficiently (Papadimitriou & Tsitsiklis, 1987).

In this paper, we aim to take a step toward the theoretical understanding of sequential decision-making with preference feedback. Specifically, we seek to understand: ($a$) *What can be learned when no assumptions are made beyond the fact that the human provides preference feedback?* This involves introducing and studying notions of dominance and optimality. ($b$) *How can we approximate preferences with a utility, making the fewest assumptions?* This requires defining a notion of *compatibility* between preference relations and utilities (Evren & Ok, 2011) and studying whether constructing a compatible utility can be done efficiently. ($c$) *How can we convert a utility to a reward function?* This includes analyzing the level of approximation and the computational tractability of the conversion.

Unlike RLHF, we will make no assumptions about the existence of an underlying reward function or the existence of a probabilistic model guiding the human preference-generation process. Our main goal is to establish a theoretical basis to design, in future works, statistically efficient algorithms for learning with preference feedback.

**Original Contributions.** The contributions of the paper are summarized as follows:

- In Section 3, we define three augmentations of the *Markov decision process without rewards* setting to include preferences, utilities and rewards.
- In Section 4, we define the notion of *compatibility* between a (partial) preorder that we use to represent preferences and a (multi-dimensional) utility function. We study the computational complexity of constructing compatible utilities. Moreover, we propose a heuristic to compute a compatible utility in polynomial time.
- In Section 5, we define the concepts of *dominance* and *optimality* for policies when only preferences are involved, discussing their computational properties, and deriving a method to verify policy dominance w.r.t. a preorder.

---

[1]A classical assumption is that the probability of one trajectory being preferred over another is proportional to some function of the difference in utility between the two (Saha et al., 2023).

[2]Intuitively, with preferences, we can only say *if* a trajectory is better than another; whereas with a utility or reward, we have to encode *how much* a trajectory is better than another.

- In Section 6, we study the problem of jointly computing a (non-Markovian) compatible utility and its (Markovian) approximation induced by rewards and we provide a bound to the distance of the induced Pareto frontiers.

Related works are reported in Section 7 and omitted proofs can be found in Appendix A.

## 2. Preliminaries

In this section, we provide the background that will be employed in the following sections.

**Notation.** Given $a, b \in \mathbb{N}$ with $a < b$, we define $[\![a]\!] := \{1, 2, \ldots, a\}$ and $[\![a, b]\!] := \{a, a+1, \ldots, b\}$. For $c \in \mathbb{R}$, we use the notation $(c)^+ := \max\{0, c\}$. Given a finite set $\mathcal{X}$, we denote as $\Delta(\mathcal{X})$ the probability simplex over $\mathcal{X}$, with $\mathcal{P}(\mathcal{X})$ its power set, and with $|\mathcal{X}|$ its cardinality. For a matrix $\mathbf{A}$, we indicate with $\|\mathbf{A}\|_\mathrm{F}$ its Frobenius norm and with $\mathbf{I}_d$ the identity matrix of order $d$.

**(Pre)Order Relations.** Let $\mathcal{X}$ be a set and $\preceq_{\mathcal{X}} \subseteq \mathcal{X} \times \mathcal{X}$ be a (binary) relation, if $(x, y) \in \preceq_{\mathcal{X}}$, we use the notation $x \preceq_{\mathcal{X}} y$. A relation $\preceq_{\mathcal{X}}$ is a *(partial) preorder* if it is: $(i)$ reflexive (i.e., $x \preceq_{\mathcal{X}} x$) and $(ii)$ transitive (i.e., $x \preceq_{\mathcal{X}} y \wedge y \preceq_{\mathcal{X}} z \Rightarrow x \preceq_{\mathcal{X}} z$). A *(partial) order* is a preorder that is $(iii)$ antisymmetric (i.e., $x \preceq_{\mathcal{X}} y \wedge y \preceq_{\mathcal{X}} x \Rightarrow x = y$). We write $x \prec_{\mathcal{X}} y$ if $x \preceq_{\mathcal{X}} y$ and not $y \preceq_{\mathcal{X}} x$. $x$ and $y$ are *incomparable*, and we denote it as $x \parallel_{\mathcal{X}} y$, if neither $x \preceq_{\mathcal{X}} y$ nor $y \preceq_{\mathcal{X}} x$; otherwise they are *comparable*. Moreover, $x$ and $y$ are *equivalent* if $x \preceq_{\mathcal{X}} y$ and $y \preceq_{\mathcal{X}} x$, and we denote it as $x \asymp_{\mathcal{X}} y$. $\asymp_{\mathcal{X}}$ is an equivalence relation that induces a partial order over the quotient set $\mathcal{X}/\asymp_{\mathcal{X}}$, i.e., $[x] \preceq_{[\mathcal{X}]/\asymp_{\mathcal{X}}} [y]$ if $x \preceq_{\mathcal{X}} y$. A (pre)order is *total* when every pair of distinct elements is comparable (i.e., $\forall x, y \in \mathcal{X} : x \preceq_{\mathcal{X}} y \vee y \preceq_{\mathcal{X}} x$). We sometimes denote total (pre)orders with the symbol $\leqslant_{\mathcal{X}}$.

**Linear Extensions, Order Dimension, and Width.** Let $\preceq_{\mathcal{X}} \in \mathcal{X} \times \mathcal{X}$ be an order relation and $\leqslant_{\mathcal{X}} \in \mathcal{X} \times \mathcal{X}$ be a total order, $\leqslant_{\mathcal{X}}$ is a *linear extension* of $\preceq_{\mathcal{X}}$ if $\preceq_{\mathcal{X}} \subseteq \leqslant_{\mathcal{X}}$ (i.e., $x \preceq_{\mathcal{X}} y \Rightarrow x \leqslant_{\mathcal{X}} y$). A set $\{\leqslant_{\mathcal{X},i}\}_{i \in [\![d]\!]}$ of total orders is a *realizer* of an order $\preceq_{\mathcal{X}}$ if $\preceq_{\mathcal{X}} = \bigcap_{i \in [\![d]\!]} \leqslant_{\mathcal{X},i}$ (which implies that all $\leqslant_{\mathcal{X},i}$ are linear extensions of $\preceq_{\mathcal{X}}$). The *order dimension* (Dushnik & Miller, 1941; Trotter, 1992) of the order $\preceq_{\mathcal{X}}$ is the least cardinality of a realizer of $\preceq_{\mathcal{X}}$, i.e., $\dim(\preceq_{\mathcal{X}}) := \min\{d \in \mathbb{N} : \exists \{\leqslant_{\mathcal{X},i}\}_{i \in [\![d]\!]} \text{ realizer of } \preceq_{\mathcal{X}}\}$. If $\preceq$ is a preorder, we define its dimension as the dimension of the partial order induced over the quotient set, i.e., $\dim(\preceq_{\mathcal{X}}) := \dim(\preceq_{\mathcal{X}/\asymp_{\mathcal{X}}})$. It is known that for $|\mathcal{X}| \geqslant 3$, computing the order dimension is NP-hard (Yannakakis, 1982; Felsner et al., 2017). Furthermore, unless NP = ZPP, there exists no polynomial-time algorithm to approximate the order dimension with a factor of $O(|\mathcal{X}|^{1-\epsilon})$, for every $\epsilon > 0$ (Chalermsook et al., 2013). An antichain (resp. chain) is a subset of $\mathcal{X}$ such that any two distinct elements are incomparable (resp. all elements are comparable). The *width* is the maximum cardinality of an antichain $\mathrm{width}(\preceq_{\mathcal{X}}) := \max\{|\mathcal{Y}| : \mathcal{Y} \subseteq \mathcal{X} \text{ s.t. } \forall x, y \in \mathcal{Y} : x \neq y \Rightarrow x \parallel_{\mathcal{X}} y\}$. It is known that $\dim(\preceq_{\mathcal{X}}) \leqslant \mathrm{width}(\preceq_{\mathcal{X}})$ (Dilworth, 1987).

**Component-wise Order.** For real vectors $\boldsymbol{v}, \boldsymbol{w} \in \mathbb{R}^d$, we define the *component-wise* (or Pareto) partial order as $\boldsymbol{v} \preceq \boldsymbol{w} \Leftrightarrow \forall i \in [\![d]\!] : v_i \leqslant w_i$. According to previous definition, we have $\boldsymbol{v} \prec \boldsymbol{w} \Leftrightarrow \forall i \in [\![d]\!] : v_i \leqslant w_i \wedge \exists j \in [\![d]\!] : v_j < w_j$.

**Sorting function.** Let $\leqslant_{\mathcal{X}}$ be a total order, a bijection $\psi_{\leqslant} : [\![|\mathcal{X}|]\!] \to \mathcal{X}$ is a *sorting function* if for every $i, j \in [\![|\mathcal{X}|]\!]$, we have $i \geqslant j \Leftrightarrow \psi_{\leqslant}(i) \leqslant_{\mathcal{X}} \psi_{\leqslant}(j)$. $\psi_{\leqslant}$ (which is unique) sorts the elements of $\mathcal{X}$ according to the total order $\leqslant_{\mathcal{X}}$. Let $f : \mathcal{X} \to \mathbb{R}$ and $\leqslant_{\mathcal{X}}$ be a total order, whenever clear from the context, we abbreviate $f(i) := f(\psi_{\leqslant_{\mathcal{X}}}(i))$.

**Markov Decision Process without Rewards.** A finite-horizon *Markov decision process without reward* (MDP\R, Abbeel & Ng, 2004) is a tuple $(\mathcal{S}, \mathcal{A}, H, p, \mu)$, where $\mathcal{S}$ and $\mathcal{A}$ are the finite ($|\mathcal{S}| =: S$ and $|\mathcal{A}| =: A$) state and action spaces, $H \in \mathbb{N}$ is the horizon, $p = (p_h)_{h \in [\![H]\!]}$ defined for every $h \in [\![H]\!]$ as $p_h : \mathcal{S} \times \mathcal{A} \to \Delta(\mathcal{S})$ is the transition model that for every state $s \in \mathcal{S}$, action $a \in \mathcal{A}$, stage $h \in [\![H]\!]$, and next state $s' \in \mathcal{S}$ provides the probability $p_h(s'|s, a)$ to reach $s'$ by playing action $a$ in state $s$ at stage $h$, and $\mu \in \Delta(\mathcal{S})$ is the initial-state distribution such that $\mu(s)$ provides the probability that the interaction starts in $s$. A *trajectory* of length $h \in [\![H]\!]$ is $\tau := (s_i, a_i)_{i \in [\![h]\!]}$, representing sequence of state-action pairs belonging to the set of trajectories $\mathcal{T}_h \subseteq (\mathcal{S} \times \mathcal{A})^h$ with cardinality $|\mathcal{T}_h| \leqslant (SA)^h$. If the length is not specified, it is assumed to be $h = H$ (i.e., $\mathcal{T} = \mathcal{T}_H$). The agent behavior is modeled with a history-dependent policy $\pi = (\pi_h)_{h \in [\![H]\!]}$ defined for every $h \in [\![H]\!]$ as $\pi_h : \mathcal{T}_{h-1} \times \mathcal{S} \to \Delta(\mathcal{A})$ that, for every trajectory $\tau \in \mathcal{T}_{h-1}$ of length $h-1$, state $s \in \mathcal{S}$, and action $a \in \mathcal{A}$, provides the probability $\pi_h(a|\tau, s)$ to play action $a$ after having observed trajectory $\tau$ and state $s$. A policy is Markovian if it depends on the current state only and, in such a case, we abbreviate with $\pi_h(a|s)$. We denote with $\Pi$ the set of history-dependent policies. A policy $\pi \in \Pi$ induces a trajectory distribution:

$$d_\pi(\tau) = \mu(s_1) \prod_{h=1}^{H} \pi_h(a_h|\tau_{h-1}, s_h) p_h(s_{h+1}|s_h, a_h), \quad (1)$$

where $\tau_l = (s_1, a_1, \ldots, s_l, a_l)$ denotes the prefix of length $l \in [\![H]\!]$ of trajectory $\tau = (s_1, a_1, \ldots, s_H, a_H)$.

## 3. Setting

In this section, we introduce three augmentations of MDP\R defined in terms of *preference* relations, *utility* function, and Markovian cumulative *reward* function.

**Preference-based MDP.** Let $\preceq_{\mathcal{T}} \subseteq \mathcal{T} \times \mathcal{T}$ be a preorder over trajectories $\mathcal{T}$. We define a *preference-based*

*Markov decision process* (PbMDP) as the tuple $\mathcal{M} = (\mathcal{S}, \mathcal{A}, H, p, \mu, \preceq_\mathcal{T})$ obtained by pairing an MDP\R with a preorder relation $\preceq_\mathcal{T}$ defining preferences over the trajectories.[3] The use of a preorder relation allows formalizing when a trajectory $\tau'$ is *preferred* over $\tau$, i.e., $\tau \preceq_\mathcal{T} \tau'$, but also accounting for both *equivalent* $\tau \asymp_\mathcal{T} \tau$ and *incomparable* $\tau \parallel_\mathcal{T} \tau'$ trajectories with $\tau, \tau' \in \mathcal{T}$. We will introduce the optimality conditions for a PbMDP in Section 5.

**Utility-based MDP.** Let $m \in \mathbb{N}$ and $\boldsymbol{u} : \mathcal{T} \to \mathbb{R}^m$ be a *multi-dimensional utility function*, i.e., a function mapping a trajectory $\tau \in \mathcal{T}$ to a vector $\boldsymbol{u}(\tau) = (u_1(\tau), \ldots, u_m(\tau))^\top$ of $m$ real numbers. A *utility-based Markov decision process* (UtilMDP) is defined as the tuple $\mathcal{M} = (\mathcal{S}, \mathcal{A}, H, p, \mu, \boldsymbol{u})$ obtained by pairing an MDP\R with a utility function $\boldsymbol{u}$. Let $\pi \in \Pi$ be a policy, its *expected utility* is defined as:

$$\boldsymbol{J}(\pi; \boldsymbol{u}) := \sum_{\tau \in \mathcal{T}} d_\pi(\tau) \boldsymbol{u}(\tau) = \langle d_\pi, \boldsymbol{u} \rangle. \tag{2}$$

Let $\pi, \pi' \in \Pi$ be two policies, we say that $\pi$ $\boldsymbol{u}$-*Pareto strictly dominates* $\pi'$ (resp. $\pi$ $\boldsymbol{u}$-*Pareto weakly dominates* $\pi'$) if $\boldsymbol{J}(\pi; \boldsymbol{u}) > \boldsymbol{J}(\pi'; \boldsymbol{u})$ (resp. $\boldsymbol{J}(\pi; \boldsymbol{u}) \geq \boldsymbol{J}(\pi'; \boldsymbol{u})$). We define the set of $\boldsymbol{u}$-*Pareto optimal* policies (i.e., the Pareto frontier) as the set of policies that are not $\boldsymbol{u}$-Pareto strictly dominated by any other policy, i.e., $\Pi^*(\boldsymbol{u}) := \{\pi \in \Pi : \neg \exists \pi' \in \Pi \text{ s.t. } \boldsymbol{J}(\pi'; \boldsymbol{u}) > \boldsymbol{J}(\pi; \boldsymbol{u})\}$. Given a utility $\boldsymbol{u}$, the $\boldsymbol{u}$-Pareto dominance induces a partial preorder relation $\preceq_{\boldsymbol{u}} \in \Pi \times \Pi$ over the policy space, of which the set of Pareto optimal policies $\Pi^*(\boldsymbol{u})$ are the maximal elements. If $m = 1$, a $u$-*optimal policy* is any policy maximizing the expected utility, i.e., $\pi^* \in \Pi^*(u) := \arg\max_{\pi \in \Pi} J(\pi; u)$.

**Reward-based MDP.** Let $m \in \mathbb{N}$ and let $\boldsymbol{r} = (\boldsymbol{r}_h)_{h \in [\![H]\!]}$, defined for every $h \in [\![H]\!]$ as $\boldsymbol{r}_h : \mathcal{S} \times \mathcal{A} \to \mathbb{R}^m$, be a *multi-dimensional reward function*, i.e., a function mapping every stage $h \in [\![H]\!]$, state $s \in \mathcal{S}$, and action $a \in \mathcal{A}$ to a vector $\boldsymbol{r}_h(s, a) = (r_{h,1}(s, a), \ldots, r_{h,m}(s, a))^\top$ of $m$ real numbers. A *(reward-based) Markov decision process* (MDP) is defined as the tuple $\mathcal{M} = (\mathcal{S}, \mathcal{A}, H, p, \mu, \boldsymbol{r})$ obtained by pairing an MDP\R with a reward function $\boldsymbol{r}$. It is always possible to define a utility from a reward by means of the *trajectory return*, defined for every $\tau = (s_1, a_1, \ldots, s_H, a_H) \in \mathcal{T}$ as:

$$\boldsymbol{u_r}(\tau) := \sum_{h=1}^{H} \boldsymbol{r}_h(s_h, a_h). \tag{3}$$

Let $\pi \in \Pi$ be a policy, its *expected return* is defined as $\boldsymbol{J}(\pi; \boldsymbol{r}) := \boldsymbol{J}(\pi; \boldsymbol{u_r})$. The concept of $\boldsymbol{r}$-Pareto dominance, the set of $\boldsymbol{r}$-Pareto optimal policies $\Pi^*(\boldsymbol{r})$, and, in the case of $m = 1$, the set of optimal policies $\Pi^*(r)$, are defined as for the UtilMDP, by means of the return utility $\boldsymbol{u_r}$. It is well-known that in MDPs there always exist (Pareto) optimal policies which are Markovian (Puterman, 2014).

---

[3]In agreement with the literature (Ok, 2002), we use *preorders* to represent the informal notion of "preference relation".

## 4. Representing Preferences with Utilities

In this section, we show how preferences can be represented using utilities. We define the notion of *compatibility* between preferences and (possibly multi-dimensional) utilities, starting with the simpler case of total preorders and, then, moving to partial preorders. We also discuss the computational aspects of constructing a compatible utility from a preorder. The content of this section will be necessary to define the notion of optimality presented in Section 5.

The use of utilities to represent preferences dates back to (Von Neumann & Morgenstern, 1947), which shows that any rational agent defines their preferences in terms of an underlying utility function. Then, (Debreu, 1954) shows the existence of a scalar utility that represents a total order. Subsequently, (Ok, 2002; Evren & Ok, 2011) extend this result by proving the existence of a multi-dimensional utility that represents a partial (pre)order relation.

**Compatible Utilities.** We start with the total preorder case.

**Definition 4.1** (Compatible Utility – Total Preorder). *Let $\leqslant_\mathcal{T}$ be a total preorder over $\mathcal{T}$ and let $u : \mathcal{T} \to \mathbb{R}$ be a scalar utility function. $u$ is* compatible *with $\leqslant_\mathcal{T}$ if for every $\tau, \tau' \in \mathcal{T}$ it holds that $\tau \leqslant_\mathcal{T} \tau' \Leftrightarrow u(\tau) \leqslant u(\tau')$.*

Thus, if $\tau <_\mathcal{T} \tau'$ (i.e., $\tau'$ is strictly preferred over $\tau$) then $u(\tau) < u(\tau')$ and if $\tau \asymp_\mathcal{T} \tau'$ (i.e., $\tau'$ and $\tau$ are equivalent) then $u(\tau) = u(\tau')$. Utilities compatible with total preorders clearly exist and a simplistic way to derive a compatible utility is to order the trajectories according to $\leqslant_\mathcal{T}$ and map each one to a real number, e.g., $u(\psi_{\leqslant_\mathcal{T}}(i)) = u(i) = |\mathcal{T}| - i$. Similarly, given a utility $u$, it is simple to derive the corresponding preorder by applying Definition 4.1. We now move to the partial preorder case, following (Ok, 2002, Equation 2).

**Definition 4.2** (Compatible Utility – Partial Preorder). *Let $\preceq_\mathcal{T}$ be a preorder over $\mathcal{T}$ and let $\boldsymbol{u} : \mathcal{T} \to \mathbb{R}^m$ with $m \in \mathbb{N}$ be a multi-dimensional utility. $\boldsymbol{u}$ is* compatible *with $\preceq_\mathcal{T}$ if for every $\tau, \tau' \in \mathcal{T}$ it holds that $\tau \preceq_\mathcal{T} \tau' \Leftrightarrow \boldsymbol{u}(\tau) \preceq \boldsymbol{u}(\tau')$.*

Some comments are in order. First, we note that, differently from Definition 4.1, we employ multi-dimensional utilities made of $m$ components. Second, we use the component-wise order of the utility to define the compatibility. Precisely, if $\tau <_\mathcal{T} \tau'$ (i.e., $\tau'$ strictly preferred over $\tau$) then $\forall i \in [\![M]\!] : u_i(\tau) \leqslant u_i(\tau')$ and $\exists j \in [\![m]\!] : u_j(\tau) < u_j(\tau')$. If, instead, $\tau \asymp_\mathcal{T} \tau'$ (i.e., $\tau$ and $\tau'$ are equivalent), we set the utilities to the same value $\forall i \in [\![m]\!] : u_i(\tau) = u_i(\tau')$. Finally, $\tau \parallel_\mathcal{T} \tau'$ (i.e., $\tau$ and $\tau'$ are incomparable) corresponds to the condition $\exists i, j \in [\![m]\!] : i \neq j \wedge u_i(\tau) > u_i(\tau') \wedge u_j(\tau) < u_j(\tau')$.

While deriving the preorder from the multi-dimensional utility can be done directly by applying Definition 4.2; differently from the total preorder case, the construction of a compatible utility from the preorder is not straightforward.

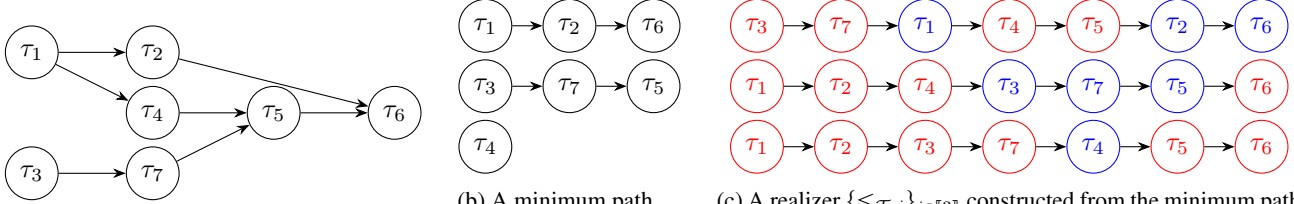

(a) DAG $\mathcal{G}$ of $\preceq_{\mathcal{T}}$.

(b) A minimum path cover $\{\mathcal{C}_i\}_{i \in [\![3]\!]}$.

(c) A realizer $\{\leqslant_{\mathcal{T},i}\}_{i \in [\![3]\!]}$ constructed from the minimum path cover $\{\mathcal{C}_i\}_{i \in [\![3]\!]}$.

*Figure 1.* Example of a partial order on the set $\mathcal{T} = \{\tau_1, \ldots, \tau_7\}$ having width $w = 3$, a minimum path cover, and a realizer.

The following result shows that the minimum value of $m$ is the order dimension of the preorder.

**Theorem 4.1.** *Let $\preceq_{\mathcal{T}} \in \mathcal{T} \times \mathcal{T}$ be a preorder over $\mathcal{T}$. Then:*

*(i) there exists a $\dim(\preceq_{\mathcal{T}})$-dimensional compatible utility;*
*(ii) no $m$-dimensional compatible utilities with $m < \dim(\preceq_{\mathcal{T}})$ exist.*

The proof of the theorem follows from the application of Definitions 4.1 and 4.2 and from the definition of order dimension. Clearly, one can define utilities with more than $\dim(\preceq_{\mathcal{T}})$ dimensions and, in any case, having fixed $m$, infinitely many compatible utilities exist (e.g., by performing translations or rescaling with positive factors). We call *minimal* a $\dim(\preceq_{\mathcal{T}})$-dimensional utility. The following result shows that computing minimal utilities is hard.

**Theorem 4.2.** *Let $\preceq_{\mathcal{T}}$ be a preorder over $\mathcal{T}$. The construction of a minimal utility $\mathbf{u}$ compatible with $\preceq_{\mathcal{T}}$ is NP-hard.*

The theorem follows from the NP-hardness of computing the order dimension. Due to the inapproximability results, it is not possible to compute in polynomial time compatible utilities with a number of dimensions $O(|\mathcal{T}|^{1-\epsilon}\dim(\preceq_{\mathcal{T}}))$ for $\epsilon > 0$ in the worst case (Chalermsook et al., 2013).

**Compatible Utility Heuristic.** We propose a method to construct a multi-dimensional utility function $\mathbf{u}$ that is compatible with $\preceq_{\mathcal{T}}$ based on dividing the problem into three phases: $(i)$ we construct a realizer $\{\leqslant_{\mathcal{T},i}\}_{i \in [\![m]\!]}$ (i.e., a set of linear extensions) of $\preceq_{\mathcal{T}}$ of size $m$ (which need not be minimal), then, $(ii)$ we construct a scalar compatible utility for each $\leqslant_{\mathcal{T},i}$ in the realizer set (which can be done in $O(|\mathcal{T}|)$ time) for every $i \in [\![m]\!]$, finally, $(iii)$ we juxtapose the scalar utilities into an $m$-dimensional utility (which can be done in $O(m)$ time).

We now introduce a tractable method for $(i)$, i.e., to derive a realizer of cardinality $w := \mathrm{width}(\preceq_{\mathcal{T}})$ given a partial order over trajectories.[4] We start by observing that $\preceq_{\mathcal{T}}$ can be represented as a *direct acyclic graph* (DAG) $\mathcal{G} = (\mathcal{T}, \mathcal{E})$, where

the set of nodes corresponds to the set of trajectories $\mathcal{T}$ and the set of edges $\mathcal{E}$ is such that its reflexive and transitive closure is the partial order $\preceq_{\mathcal{T}}$.[5] We now solve a *minimum path cover* (MPC) problem to obtain a set of $w$ chains (i.e., paths in the graph) that covers all the trajectories (i.e., all the nodes). Caceres et al. (2022) proposes an algorithm that runs in $O(w^2|\mathcal{T}| + |\mathcal{E}|)$. Letting $\{\mathcal{C}_i\}_{i \in [\![w]\!]}$ represent the set of chains (i.e., sequence of nodes), we now derive a realizer set $\{\leqslant_{\mathcal{T},i}\}_{i \in [\![w]\!]}$. This is done by extending each chain $\mathcal{C}_i$ with $i \in [\![w]\!]$ to obtain the linear extension $\leqslant_{\mathcal{T},i}$ as follows: for every $\tau_1, \tau_2 \in \mathcal{T}$, if $\tau_1$ and $\tau_2$ are incomparable in $\preceq_{\mathcal{T}}$ (i.e., $\tau_1 \| _{\mathcal{T}} \tau_2$) and $\tau_2 \in \mathcal{C}_i$, then $\tau_1 \leqslant_{\mathcal{T},i} \tau_2$. This procedure has cost of $O(|\mathcal{T}|^2)$. Overall, we can compute *a* realizer of $\preceq_{\mathcal{T}}$ with cardinality $w$ in at most $O(|\mathcal{T}|(|\mathcal{T}| + w^2))$, having observed that $|\mathcal{E}| \leqslant w|\mathcal{T}|$ (Kritikakis & Tollis, 2022). An example of this procedure is reported in Figure 1.

Given Definitions 4.1 and 4.2, every UtilMDP can be mapped to exactly one PbMDP defined with the preorder $\preceq_{\mathcal{T}}$ unambiguously constructed from the utility $\mathbf{u}$, while a PbMDP can be mapped to multiple (infinitely many) UtilMDPs with any utility $\mathbf{u}$ compatible with the preorder $\preceq_{\mathcal{T}}$. This observation motivates the need for evaluating optimality and dominance directly w.r.t. the preference relation.

## 5. Dominance and Optimality with Preferences

In this section, we introduce the novel concepts of *dominance* and *optimality* for policies defined by means of the preorder $\preceq_{\mathcal{T}}$, and we discuss their computational properties. Similarly to UtilMDPs and MDPs, where (possibly multi-dimensional) utilities or rewards are present, we aim to characterize the target when solving a PbMDP, i.e., a notion of a non-dominated set of policies. However, unlike UtilMDPs and MDPs, PbMDPs lack a numerical signal.

From now on, we only consider the case in which $\preceq_{\mathcal{T}}$ is an order. Indeed, if $\preceq_{\mathcal{T}}$ is a preorder, we can consider the order induced over the quotient $\mathcal{T}/\asymp_{\mathcal{T}}$, observing that equivalent trajectories correspond to the same utility value.

**Dominance for Total Orders.** As discussed in Section 4,

---

[4]We consider only the case in which we have an order. Indeed, if we have a preorder, we can consider the order induced over the quotient set by the equivalence relation $\asymp_{\mathcal{T}}$, as for equivalent trajectories, we are forced to set the same value of the utility.

[5]Formally, $\mathcal{E} \subseteq \mathcal{T} \times \mathcal{T}$ is the *cover relation* induced by the partial order $\preceq_{\mathcal{T}}$ (Knuth, 2013).

for every order $\leqslant_{\mathcal{T}}$, there exist infinitely many compatible utilities. However, the Pareto optimality of a policy $\pi \in \Pi$ w.r.t. a certain compatible utility $u$ does not necessarily guarantee its Pareto optimality w.r.t. another compatible utility $u'$, as shown in the following example.

**Example 1.** *This holds even for scalar utilities. Let $\mathcal{T} = \{\tau_1, \tau_2, \tau_3\}$ and the total order $\leqslant_{\mathcal{T}}$ be defined as:*

$$\tau_3 <_{\mathcal{T}} \tau_2 <_{\mathcal{T}} \tau_1. \tag{4}$$

*Let $\Pi = \{\pi, \pi'\}$ be the policy space with the corresponding trajectory distributions $d_\pi = (0.5, 0.5, 0)^\top$ and $d_{\pi'} = (0.8, 0, 0.2)^\top$. Consider the utilities $u_1 = (4, 2, 0)^\top$ and $u_2 = (4, 2, -2)^\top$ both compatible with $\leqslant_{\mathcal{T}}$. We have:*

$$J(\pi; u_1) = J(\pi; u_2) = 3, \tag{5}$$

$$J(\pi'; u_1) = 3.2, \quad J(\pi'; u_2) = 2.8. \tag{6}$$

*Thus, $\pi'$ $u_1$-(Pareto) dominates $\pi$ and $\pi$ $u_2$-(Pareto) dominates $\pi'$.*

For this reason, we propose defining dominance between policies considering *all compatible utilities*. This ensures that if a policy $\pi$ dominates another policy $\pi'$ (in the sense defined below), then $\pi$ Pareto dominates $\pi'$ w.r.t. all compatible utilities. Let us begin with the case of total orders.

**Definition 5.1** (Policy Dominance – Total Order). *Let $\leqslant_{\mathcal{T}}$ be a total order over $\mathcal{T}$, and let $\pi, \pi' \in \Pi$ be two policies. $\pi$ $\leqslant_{\mathcal{T}}$-strictly dominates $\pi'$, denoted as $\pi' <_\Pi \pi$ if, for every utility $u : \mathcal{T} \to \mathbb{R}$ compatible with $\leqslant_{\mathcal{T}}$, we have:*

$$J(\pi; u) - J(\pi'; u) = \langle d_\pi - d_{\pi'}, u \rangle > 0.$$

*If the inequality holds with $\geqslant$, we say that $\pi$ $\leqslant_{\mathcal{T}}$-weakly dominates $\pi'$, denoted as $\pi' \leqslant_\Pi \pi$.*

Since we are considering total orders and, consequently, scalar utilities, we require that $\pi$ yields a strictly better expected utility $J(\pi; u)$ compared to that $J(\pi'; u)$ of $\pi'$, evaluated *under any compatible utility*. Note that $\leqslant_\Pi \in \Pi \times \Pi$ is a partial preorder over the space of policies $\Pi$. Indeed, even if the order $\leqslant_{\mathcal{T}}$ is *total*, the induced preorder $\leqslant_\Pi$ can be *partial*, as illustrated below.

**Example 2.** *Let $\mathcal{T} = \{\tau_1, \tau_2, \tau_3, \tau_4\}$ be a trajectory space. Consider the following total order $\leqslant_{\mathcal{T}}$:*

$$\tau_4 <_{\mathcal{T}} \tau_3 <_{\mathcal{T}} \tau_2 <_{\mathcal{T}} \tau_1. \tag{7}$$

*Let $\pi, \pi' \in \Pi$ be two policies with trajectory distributions $d_\pi = (0.4, 0.3, 0.1, 0.2)^\top$ and $d_{\pi'} = (0.3, 0.2, 0.4, 0.1)^\top$. Now, let $u_1 = (4, 3, 2, 1)^\top$ and $u_2 = (10, 9, 8, 1)^\top$ be two scalar utilities both compatible with $\leqslant_{\mathcal{T}}$. Thus, to determine whether $\pi$ dominates $\pi'$, we need to verify if the condition of Definition 5.1 holds for both utilities: $\langle d_\pi - d_{\pi'}, u_1 \rangle = 0.2$ and $\langle d_\pi - d_{\pi'}, u_2 \rangle = -0.4$. Thus, $\pi$ does not dominate $\pi'$ and vice versa (i.e., $\pi' \|_\Pi \pi$), showing that $\leqslant_\Pi$ is partial.*

Definition 5.1 requires testing the condition "for every compatible utility" which is clearly infeasible. We can easily overcome this issue, as shown in the following result.

**Theorem 5.1.** *Let $\leqslant_{\mathcal{T}}$ be a total order over $\mathcal{T}$, and let $\pi, \pi' \in \Pi$ be two policies. $\pi$ $\leqslant_{\mathcal{T}}$-weakly dominates $\pi'$ if and only if it holds that:*

$$\forall n \in [\![|\mathcal{T}|]\!] : \sum_{i=1}^n (d_\pi(i) - d_{\pi'}(i)) \geqslant 0. \tag{8}$$

*Furthermore, $\pi$ $\leqslant_{\mathcal{T}}$-strictly dominates $\pi'$ if and only if, in addition to the above, it holds that:*

$$\exists n' \in [\![|\mathcal{T}|]\!] : \sum_{i=1}^{n'} (d_\pi(i) - d_{\pi'}(i)) > 0. \tag{9}$$

The proof is reported in Appendix A. To give an interpretation to the condition in Equation (8), consider the vectors $\boldsymbol{d_\pi} = (d_\pi(1), \ldots, d_\pi(|\mathcal{T}|))^\top$ and $\boldsymbol{d_{\pi'}} = (d_{\pi'}(1), \ldots, d_{\pi'}(|\mathcal{T}|))^\top$ of the trajectory probabilities sorted in non-increasing order (from the most preferred to the least preferred trajectory) according to the total order $\leqslant_{\mathcal{T}}$. Equation (8) prescribes that the vectors of the *cumulative sums* $\mathbf{C}\boldsymbol{d_\pi}$ and $\mathbf{C}\boldsymbol{d_{\pi'}}$ of the trajectory probabilities to satisfy $\mathbf{C}\boldsymbol{d_\pi} \geq \mathbf{C}\boldsymbol{d_{\pi'}}$ in the sense of the component-wise order, where $\mathbf{C}$ is a lower triangular matrix of all 1s. Thus, we have reduced the problem of assessing the dominance between policies ($\pi' \leqslant_\Pi \pi$) to the problem of assessing dominance between real vectors ($\mathbf{C}\boldsymbol{d_{\pi'}} \leq \mathbf{C}\boldsymbol{d_\pi}$). An immediate intuitive consequence is that for the most preferred trajectory, we have $d_\pi(1) \geqslant d_{\pi'}(1)$, and for the least preferred trajectory, we have $d_\pi(|\mathcal{T}|) \leqslant d_{\pi'}(|\mathcal{T}|)$. The computational complexity of verifying the condition of Equation (8) is $O(|\mathcal{T}|)$.

**Dominance for Partial Orders.** Moving from total to partial orders, we directly generalize Definition 5.1 to the case of compatible (multi-dimensional) utilities.

**Definition 5.2** (Policy Dominance – Partial Order). *Let $\preceq_{\mathcal{T}}$ be an order over $\mathcal{T}$, and let $\pi, \pi' \in \Pi$ be two policies. $\pi$ $\preceq_{\mathcal{T}}$-strictly dominates $\pi'$, denoted as $\pi' <_\Pi \pi$ if, for every utility $\boldsymbol{u} : \mathcal{T} \to \mathbb{R}$ compatible with $\preceq_{\mathcal{T}}$, it holds that:*

$$\boldsymbol{J}(\pi; \boldsymbol{u}) - \boldsymbol{J}(\pi'; \boldsymbol{u}) = \langle d_\pi - d_{\pi'}, \boldsymbol{u} \rangle > \mathbf{0}.$$

*If the inequality holds with $\succeq$, we say that $\pi$ $\preceq_{\mathcal{T}}$-weakly dominates $\pi'$, denoted as $\pi' \preceq_\Pi \pi$.*

Thus, we require that policy $\pi$ $\boldsymbol{u}$-Pareto dominates $\pi'$ *under any compatible utility $\boldsymbol{u}$.* As for the case of total orders, $\preceq_\Pi \in \Pi \times \Pi$ represents a partial preorder over the space of policies. The following result shows that Definition 5.2, i.e., dominance between policies w.r.t. a partial order $\preceq_{\mathcal{T}}$, can be equivalently stated by requiring that dominance holds for all the linear extensions (i.e., total orders), according to Definition 5.1, for every realizer $\{\leqslant_{\mathcal{T}, i}\}_{i \in [\![m]\!]}$ of $\preceq_{\mathcal{T}}$.

**Theorem 5.2.** *Let $\leq_{\mathcal{T}}$ be a partial order over $\mathcal{T}$ and let $\pi, \pi' \in \Pi$ be two policies. $\pi \leq_{\mathcal{T}}$-weakly dominates $\pi'$ if and only if, for every realizer $\{\leqslant_{\mathcal{T},i}\}_{i \in [\![m]\!]}$ with $m \in \mathbb{N}$ of $\leq_{\mathcal{T}}$, it holds that:*

$$\forall i \in [\![m]\!]: \quad \pi' \leqslant_{\Pi,i} \pi,$$

*where $\pi' \leqslant_{\Pi,i} \pi$ (resp. $\pi' <_{\Pi,i} \pi$) denotes that $\pi$ weakly (resp. strictly) $\leqslant_{\mathcal{T},i}$-dominates $\pi'$ (Definition 5.1) w.r.t. the $i$-th total order in the realizer of $\leq_{\mathcal{T}}$. Furthermore, $\pi \leq_{\mathcal{T}}$-strictly dominates $\pi'$ if and only if, in addition to the above, it holds that:*

$$\exists j \in [\![m]\!]: \quad \pi' <_{\Pi,j} \pi. \tag{10}$$

Thus, we have reduced the problem of assessing the dominance for partial orders to assessing the dominance of a number of total orders. By a simple application of Theorem 5.1, we can state the following equivalent condition.

**Theorem 5.3.** *Let $\leq_{\mathcal{T}}$ be a partial order over $\mathcal{T}$ and let $\pi, \pi' \in \Pi$ be two policies. $\pi \leq_{\mathcal{T}}$-weakly dominates $\pi'$ if and only if, for every linear extension $\leqslant_{\mathcal{T}}$ of $\leq_{\mathcal{T}}$, it holds that:*

$$\forall n \in [\![|\mathcal{T}|]\!]: \sum_{i=1}^{n} (d_{\pi}(\psi_{\leqslant_{\mathcal{T}}}(i)) - d_{\pi'}(\psi_{\leqslant_{\mathcal{T}}}(i))) \geqslant 0. \tag{11}$$

*$\pi \leq_{\mathcal{T}}$-strictly dominates $\pi'$ if and only if, in addition to the above, there exists a linear extension $\leqslant'_{\mathcal{T}}$ of $\leq_{\mathcal{T}}$ such that:*

$$\exists n \in [\![|\mathcal{T}|]\!]: \sum_{i=1}^{n} (d_{\pi}(\psi_{\leqslant'_{\mathcal{T}}}(i)) - d_{\pi'}(\psi_{\leqslant'_{\mathcal{T}}}(i))) > 0. \tag{12}$$

Although it resembles Theorem 5.1 for total orders, Theorem 5.3 cannot be leveraged to derive an efficient algorithm. Indeed, a trivial application would require to enumerate all linear extensions that, in the worst case, are $|\mathcal{T}|!$. We are currently unable to provide a polynomial-time algorithm to assess policy dominance for partial orders but we conjecture that the problem is computationally hard.

**Optimality.** We now define a notion of optimality for policies in terms of the preference relation. Following the same ideas as for Pareto-optimal policies, we call a policy optimal w.r.t. an order $\leq_{\mathcal{T}}$ if there exists no other policy that strictly dominates it.

**Definition 5.3** (Optimality). *Let $\leq_{\mathcal{T}}$ be a partial order over $\mathcal{T}$. $\pi^* \in \Pi$ is $\leq_{\mathcal{T}}$-optimal if it is not $\leq_{\mathcal{T}}$-strictly dominated by any other policy. We denote the set of $\leq_{\mathcal{T}}$-optimal policies as:*

$$\Pi^*(\leq_{\mathcal{T}}) := \{\pi \in \Pi : \neg \exists \pi' \in \Pi \text{ s.t. } \pi <_{\Pi} \pi'\}.$$

# 6. From (Non-Markovian) Utility to Markovian Reward

In this section, we study the problem of approximating a (non-Markovian) compatible utility with a (Markovian) reward and discuss the approximation error.

**Total Order Case.** Consider a total order $\leqslant_{\mathcal{T}}$ over $|\mathcal{T}|$ trajectories that can be represented by a scalar compatible utility $u \in \mathbb{R}^{|\mathcal{T}|}$, as in Definition 4.1. We can arbitrarily choose the values of $u(i)$ so that for every $i, j \in [\![|\mathcal{T}|]\!]$ such that $i < j$ we have $u(i) \leqslant u(j) - \varepsilon$ where $\varepsilon > 0$ represents the *minimum utility gap* between two trajectories. We want to find a reward vector $r \in \mathbb{R}^{SAH}$, which best represents the compatible utility vector. To this end, we jointly optimize the choice of utility $u$ and reward $r$ to minimize the error due to the limited expressive power of the reward w.r.t. the utility, by means of the following *quadratic program* (QP):

$$\eta^* := \min_{u \in \mathbb{R}^{|\mathcal{T}|}, r \in \mathbb{R}^{SAH}} \|u - \mathbf{B}r\|_2^2$$
$$\text{s.t.} \quad u(\psi_{\leqslant_{\mathcal{T}}}(i+1)) \leqslant u(\psi_{\leqslant_{\mathcal{T}}}(i)) - \varepsilon,$$
$$\forall i \in [\![|\mathcal{T}| - 1]\!]$$
$$u(\psi_{\leqslant_{\mathcal{T}}}(|\mathcal{T}|)) = 0$$
$$u(\psi_{\leqslant_{\mathcal{T}}}(1)) = 1$$

where $\mathbf{B} \in \{0,1\}^{|\mathcal{T}| \times SAH}$ is a binary matrix encoding, for every trajectory, which stages, states, and actions are involved in it (the order in which we design this matrix will influence only the order the elements in the reward vector).[6] The constraints on $u(1)$ and $u(|\mathcal{T}|)$ just set the scale of the utilities and the ones proposed above are an arbitrary valid choice. We can easily eliminate the variable $r$ by observing that it is not involved in any constraints, and solve the least-squares problem in closed form, obtaining $r = (\mathbf{B}^\top \mathbf{B})^{-1} \mathbf{B}^\top u$.[7] Thus, by defining $\mathbf{A} := \mathbf{I}_{|\mathcal{T}|} - \mathbf{B}(\mathbf{B}^\top \mathbf{B})^{-1} \mathbf{B}^\top$, the objective function becomes $\|\mathbf{A}u\|_2^2 = u^\top \mathbf{A}^\top \mathbf{A} u$, leading to a QP with $|\mathcal{T}|$ variables, a quadratic (convex) objective, and $|\mathcal{T}| + 1$ linear constraints, that can be solved using convenient convex optimization tools (Boyd & Vandenberghe, 2004).

**Partial Order Case.** The same rationale can be applied to partial orders $\leq_{\mathcal{T}}$ by considering a realizer $\{\leqslant_{\mathcal{T},j}\}_{j \in [\![m]\!]}$ and a compatible $m$-dimensional utility $\boldsymbol{u} \in \mathbb{R}^{|\mathcal{T}| \times m}$ (also switching the Euclidean norm with the Frobenious norm):

$$\eta^* := \min_{\boldsymbol{u} \in \mathbb{R}^{|\mathcal{T}| \times m}} \|\mathbf{A}\boldsymbol{u}\|_{\mathrm{F}}^2 \tag{13}$$
$$\text{s.t.} \quad u_j(\psi_{\leqslant_{\mathcal{T},j}}(i+1)) \leqslant u_j(\psi_{\leqslant_{\mathcal{T},j}}(i)) - \varepsilon,$$
$$\forall i \in [\![|\mathcal{T}| - 1]\!], \ j \in [\![m]\!]$$
$$u_j(\psi_{\leqslant_{\mathcal{T},j}}(|\mathcal{T}|)) = 0, \qquad \forall j \in [\![m]\!]$$
$$u_j(\psi_{\leqslant_{\mathcal{T},j}}(1)) = 1, \qquad \forall j \in [\![m]\!]$$

Also in this case we are in the presence of a QP with $m|\mathcal{T}|$ variables and $m(|\mathcal{T}| + 1)$ linear constraints.

---

[6] Formally, let $\tau = (s_1, a_1, \ldots, s_H, a_H) \in \mathcal{T}$, we have that $\mathbf{B}(\tau, (s_l, a_l, l)) = 1$ for every $l \in [\![H]\!]$ and all other components of row $\tau$ are equal to 0.

[7] The choice of the set of all trajectories $|\mathcal{T}|$ makes $\mathbf{B}$ full rank, thus ensuring that $\mathbf{B}^\top \mathbf{B}$ admits an inverse.

**Approximation Error.** When the partial order can be indeed represented via Markovian rewards, then the QP presented above returns a value of the objective function $\eta^* = 0$, otherwise, it returns $\eta^* > 0$. In the opposite case, the Markovian reward yields an approximated utility $\widehat{\boldsymbol{u}} = \boldsymbol{u_r}$, that will induce a certain set $\Pi^*(\widehat{\boldsymbol{u}}) \subseteq \Pi$ of $\widehat{\boldsymbol{u}}$-Pareto optimal policies, whereas $\boldsymbol{u}$ will yield another set $\Pi^*(\boldsymbol{u}) \subseteq \Pi$ of $\boldsymbol{u}$-Pareto optimal policies. We now propose to evaluate the dissimilarity between the two sets of policies with the following index:

$$\mathcal{L}(\boldsymbol{u}, \widehat{\boldsymbol{u}}) := \max \left\{ \sup_{\pi \in \Pi^*(\boldsymbol{u})} \inf_{\widehat{\pi} \in \Pi^*(\widehat{\boldsymbol{u}})} \Delta J^+(\pi, \widehat{\pi}, \boldsymbol{u}), \right.$$
$$\left. \sup_{\widehat{\pi} \in \Pi^*(\widehat{\boldsymbol{u}})} \inf_{\pi \in \Pi^*(\boldsymbol{u})} \Delta J^+(\widehat{\pi}, \pi, \widehat{\boldsymbol{u}}) \right\},$$

where:

$$\Delta J^+(\pi, \widehat{\pi}, \boldsymbol{u}) := \sum_{j \in [\![m]\!]} \left( J(\pi, u_j) - J(\widehat{\pi}, u_j) \right)^+. \quad (14)$$

This index is designed to account only for performance losses when we move from a $\boldsymbol{u}$-Pareto optimal policy $\pi$ to a $\widehat{\boldsymbol{u}}$-Pareto optimal policy $\widehat{\pi}$ and does not allow for compensations when $\widehat{\pi}$ better optimizes some dimensions of $\boldsymbol{u}$ w.r.t. the Pareto optimal policy $\pi$. The presence of the infimum ensures picking the policy $\widehat{\pi}$ in the Pareto frontier of $\widehat{\boldsymbol{u}}$ "closest" to $\pi$, while the supremum forces the worst-case choice of $\pi$. Analogous reasoning holds for the second argument of the max by reversing the roles of $\pi$ and $\widehat{\pi}$. In the following theorem, we upper bound the performance loss due to the Markovian approximation.

**Theorem 6.1.** *Let* $\boldsymbol{u}, \widehat{\boldsymbol{u}} : \mathcal{T} \to \mathbb{R}^m$ *be two $m$-dimensional utilities functions such that* $\|\boldsymbol{u} - \widehat{\boldsymbol{u}}\|_{\mathrm{F}}^2 \leqslant \eta^*$. *Then, it holds that* $\mathcal{L}(\boldsymbol{u}, \widehat{\boldsymbol{u}}) \leqslant 2\sqrt{m\eta^*}$.

It is worth noting that this result holds for arbitrary pairs of utilities, not necessarily derived with the QP presented above. We can trivially verify that in the case of a total preorder, the difference in performance is bounded by $2\sqrt{\eta^*}$.

# 7. Related Works

We summarize the relevant literature, focusing on feedback types, learning from preferences, and results on bandits.

**Types of Feedback.** PbRL and RLHF approaches have been studied combined with several types of feedback. Kaufmann et al. (2023) report and analyze several classes of feedback, presenting a trade-off in terms of how the complexity is distributed between the human expert (i.e., difficulty of providing a feedback) and the agent (i.e., difficulty of learning given the feedback). In our framework, we consider only feedback over trajectories, the most common one, while allowing for non-Markovianity in the implicit evaluation of the expert. Asking for a preference among a set of objects

(i.e., the type of feedback we consider in this work) is also referred to as *comparison* feedback. Comparison feedback first appeared in the literature in terms of feedback over individual state-action pairs (Cheng et al., 2011; Fürnkranz et al., 2012), and was later extended to reward learning tasks (Christiano et al., 2017; Ibarz et al., 2018).

**Learning from Preferences.** Our setting has connections with both PbRL and RLHF. Wirth et al. (2017) propose the *Markov decision process with preferences* (MDPP) setting, aiming at unifying some of the existing PbRL results under a common framework. MDPPs employ a stochastic preference generation process. Although this is a relevant scenario when learning a policy given a set of binary preferences, it deviates from the objective of studying the computational complexity of the problem, thus, motivating the need to define our PbMDPs where the preferences are deterministic. Moreover, MDPPs define preferences between trajectories in terms of the likelihood of them being generated by a given policy. This assumption, although sensible w.r.t. the goal of the authors, is stronger than what is required in this work that simply considers general preorders. Wirth et al. (2017) and Kaufmann et al. (2023) survey several PbRL and RLHF approaches, ranging in methodology from direct policy learning (Wilson et al., 2012; Rafailov et al., 2024), to learning a utility (Akrour et al., 2012), to learning a reward function (Zucker et al., 2010; Christiano et al., 2017), all under the probabilistic preference assumption.

**Preference-Based Multi-Armed Bandits.** Several *multi-armed bandit* (MAB, Lattimore & Szepesvári, 2020) settings share some aspects with PbRL. For example, *dueling bandits* (DBs, Yue et al., 2012) are the preference-based version of MABs, and can be interpreted as the one-state version of PbRL. DBs can allow for non-order relations among arms (see, e.g., Zoghi et al., 2015). Xu et al. (2020) employ a DB-based subroutine in their PbRL algorithm, and demonstrate the existence of MDPs with non-transitive preferences between trajectories, leading to the absence of a unique optimal policy. This scenario, however, is out of the scope of this work, as removing the assumption of a (partial) preorder would change the basis of the analysis, with a notable loss of the properties presented in this paper. A different example is (Azar et al., 2024), in which the authors define the problem of learning from human feedback as an *offline contextual bandit* (Lu et al., 2010) problem. We refer the interested reader to (Busa-Fekete & Hüllermeier, 2014) for a detailed survey of preference-based learning in MABs.

# 8. Discussion and Conclusions

In this work, we defined the PbMDP setting, obtained by extending an MDP\R with a (partial) preorder over trajectories, and compared it with UtilMDPs and MDPs. We

defined the notion of utility-preference compatibility and discussed the computational issues in constructing them. Then, we defined the concepts of policy dominance, accounting for the fact that the true underlying utility function is unknown. Finally, we discussed the need to move from utilities to Markovian rewards, providing a QP optimization problem to compute the reward values, and quantifying the approximation error.

**Future Works.** The computational limitations presented in the paper suggest the need for less demanding notions of dominance when preferences are concerned. Furthermore, our work does not tackle the statistical complexity of learning with preference feedback. Future works should address these issues. Specifically, it would be interesting to investigate less demanding notions of dominance that consider, e.g., a subset of *all compatible utilities*, and compare them with the one presented in this paper from the computational perspective. Moreover, in realistic scenarios, the preference relation is not given and should be learned from samples. Future studies could define methodologies to address both the preference elicitation problem (see, e.g., Wilde et al., 2018), and the uncertainty in the preference generation process. One such natural extension is to study the statistical complexity of a multi-objective problem in terms of ($i$) the uncertainty due to a partial coverage of the preorder relation and ($ii$) the error due to the approximation.

## Impact Statement

This paper presents work whose goal is to advance the field of Machine Learning. There are many potential societal consequences of our work, none which we feel must be specifically highlighted here.

## Acknowledgments

Funded by the European Union – Next Generation EU within the project NRPP M4C2, Investment 1.3 DD. 341 – 15 March 2022 – FAIR – Future Artificial Intelligence Research – Spoke 4 – PE00000013 – D53C22002380006.

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

# A. Omitted Proofs

**Theorem 4.1.** *Let $\leq_{\mathcal{T}} \in \mathcal{T} \times \mathcal{T}$ be a preorder over $\mathcal{T}$. Then:*

*(i) there exists a $\dim(\leq_{\mathcal{T}})$-dimensional compatible utility;*
*(ii) no $m$-dimensional compatible utilities with $m < \dim(\leq_{\mathcal{T}})$ exist.*

*Proof.* We limit the proof for the case in which we have an order. Indeed, if we have a preorder, we can consider the order induced over the quotient set by the equivalence relation $\asymp_{\mathcal{T}}$, as for equivalent trajectories we are forced to set the same value of the utility. Let us start with $(i)$. We show the existence of a compatible $\dim(\leq_{\mathcal{T}})$-dimensional utility. Let $D = \dim(\leq_{\mathcal{T}})$, for notational convenience. To this end, we know that there exists a set $\{\leqslant_{\mathcal{T},i}\}_{i=1}^{D}$ of $D$ total orders such that $\leq_{\mathcal{T}} = \bigcap_{i=1}^{D} \leqslant_{\mathcal{T},i}$, i.e., $\tau \leq_{\mathcal{T}} \tau' \Leftrightarrow \forall i \in [\![D]\!] : \tau \leqslant_{\mathcal{T},i} \tau'$. Since for total orders, compatible utilities exist, let us consider $u_i : \mathcal{T} \to \mathbb{R}$, compatible with $\leqslant_{\mathcal{T},i}$ for every $i \in [\![D]\!]$. Let us now construct the $D$-dimensional utility $\boldsymbol{u} = (u_1, \dots, u_D)^{\top}$. We show that $\boldsymbol{u}$ is compatible with the preorder $\leq_{\mathcal{T}}$. Let $\tau, \tau' \in \mathcal{T}$, we have:

$$\boldsymbol{u}(\tau) \preceq \boldsymbol{u}(\tau') \Leftrightarrow \forall i \in [\![D]\!] : u_i(\tau) \leqslant u_i(\tau') \tag{15}$$

$$\Leftrightarrow \forall i \in [\![D]\!] : \tau \leqslant_{\mathcal{T},i} \tau' \tag{16}$$

$$\Leftrightarrow \tau \leq_{\mathcal{T}} \tau', \tag{17}$$

where line (16) follows from the compatibilities of the scalar utilities $u_i$ with the corresponding $\leqslant_{\mathcal{T},i}$ and line (17) follows from the construction of the partial preorder from the intersection of total preorders. For $(ii)$, by contradiction, suppose there exists an $m$-dimensional compatible utility $\boldsymbol{u} = (u_1, \dots, u_m)^{\top}$ with $m < D$. Let $\{\leqslant_{\mathcal{T},i}\}_{i=1}^{m}$ be the set of $m$ total orders induced by $u_1, \dots, u_m$, which is unique. We now show that $\leq_{\mathcal{T}} = \bigcap_{i=1}^{m} \leqslant_{\mathcal{T},i}$ contradicting the definition of order dimension. Let $\tau, \tau' \in \mathcal{T}$, we have:

$$\tau \leq_{\mathcal{T}} \tau' \Leftrightarrow \boldsymbol{u}(\tau) \preceq \boldsymbol{u}(\tau') \tag{18}$$

$$\Leftrightarrow \forall i \in [\![m]\!] : u_i(\tau) \leqslant u_i(\tau') \tag{19}$$

$$\Leftrightarrow \forall i \in [\![m]\!] : \tau \leqslant_{\mathcal{T},i} \tau', \tag{20}$$

where line (18) follows from the compatibility of the multi-dimensional utility and line (20) follows from the compatibility of the scalar utilities. $\qquad\square$

**Theorem 4.2.** *Let $\leq_{\mathcal{T}}$ be a preorder over $\mathcal{T}$. The construction of a minimal utility $\boldsymbol{u}$ compatible with $\leq_{\mathcal{T}}$ is NP-hard.*

*Proof.* We restrict to the case of orders. We reduce from the problem of deciding whether the order dimension of an order is $\geqslant k$ which is known to be NP-hard (Yannakakis, 1982; Felsner et al., 2017).

**Decision Problems.**

ORDER DIMENSION (OD): given an order $\leq \subseteq \mathcal{X} \times \mathcal{X}$ and a natural number $k \in \mathbb{N}$, YES if the order dimension is $\leqslant k$.

MINIMAL UTILITY (MU): given an order $\leq \subseteq \mathcal{X} \times \mathcal{X}$ and a natural number $k \in \mathbb{N}$, YES if a minimal compatible utility has dimensionality $\leqslant k$.

**Reduction.** We show that OD $\leqslant_p$ MU ($\leqslant_p$ denotes a Karp's reduction). The instance of MU is the same as for OD. It is trivial to show that the order dimension is $\leqslant k$ if and only if a minimal compatible utility has dimensionality $\leqslant k$. $\qquad\square$

**Theorem 5.1.** *Let $\leqslant_{\mathcal{T}}$ be a total order over $\mathcal{T}$, and let $\pi, \pi' \in \Pi$ be two policies. $\pi \leqslant_{\mathcal{T}}$-weakly dominates $\pi'$ if and only if it holds that:*

$$\forall n \in [\![|\mathcal{T}|]\!] : \sum_{i=1}^{n} (d_\pi(i) - d_{\pi'}(i)) \geqslant 0. \tag{8}$$

*Furthermore, $\pi \leqslant_{\mathcal{T}}$-strictly dominates $\pi'$ if and only if, in addition to the above, it holds that:*

$$\exists n' \in [\![|\mathcal{T}|]\!] : \sum_{i=1}^{n'} (d_\pi(i) - d_{\pi'}(i)) > 0. \tag{9}$$

*Proof.* We prove the first statement, as the second one can be proved analogously.

**If.** We start showing that:

$$\pi' \leqslant_\Pi \pi \quad \Rightarrow \quad \min_{n \in [\![|\mathcal{T}|]\!]} \sum_{i=1}^{n} (d_\pi(i) - d_{\pi'}(i)) \geqslant 0$$

By contradiction, suppose the following condition to hold:

$$\exists n^* \in [\![|\mathcal{T}|]\!] : \sum_{i=1}^{n^*} (d_\pi(i) - d_{\pi'}(i)) < 0 \wedge \inf_{u \text{ compatible with } \leqslant_\mathcal{T}} \langle d_\pi - d_{\pi'}, u \rangle \geqslant 0.$$

Define the utility function $\widetilde{u}$ defined as:

$$\widetilde{u}(i) = \begin{cases} M & \text{if } i \leqslant n^*, \\ 0 & \text{if } i > n^*, \end{cases}$$

for some $M > 0$. We observe that $\widetilde{u}$ is compatible with $\leqslant_\mathcal{T}$. Then, we can write:

$$\sum_{i=1}^{|\mathcal{T}|} \widetilde{u}(i) (d_\pi(i) - d_{\pi'}(i)) = \sum_{i=1}^{n^*} \widetilde{u}(i) (d_\pi(i) - d_{\pi'}(i)) + \sum_{i=n^*+1}^{|\mathcal{T}|} \widetilde{u}(i) (d_\pi(i) - d_{\pi'}(i))$$

$$= M \sum_{i=1}^{n^*} (d_\pi(i) - d_{\pi'}(i))$$

$$< 0,$$

where the last inequality holds under condition $(i)$, which is absurd.

**Only if.** Let us now prove that:

$$\min_{n \in [\![|\mathcal{T}|]\!]} \sum_{i=1}^{n} (d_\pi(i) - d_{\pi'}(i)) \geqslant 0 \quad \Rightarrow \quad \pi' \leqslant_\Pi \pi. \tag{21}$$

The LHS of Equation (21) implies that, for every $n^* \in [\![|\mathcal{T}|]\!]$, it holds that:

$$\sum_{i=1}^{n^*} (d_\pi(i) - d_{\pi'}(i)) \geqslant 0, \tag{22}$$

and consequently, that the following holds as well:

$$\sum_{i=n^*+1}^{|\mathcal{T}|} (d_\pi(i) - d_{\pi'}(i)) < 0, \tag{23}$$

since, by definition of the policy occupancy, it holds that:

$$\sum_{i=1}^{|\mathcal{T}|} (d_\pi(i) - d_{\pi'}(i)) = 0. \tag{24}$$

Let $u$ be a compatible utility function, and let $m \in [\![|\mathcal{T}|]\!]$ be the index such that:

$$\begin{cases} u(i) \geqslant 0 & \text{if } i \leqslant m, \\ u(i) < 0 & \text{if } i > m. \end{cases}$$

Then, we can rewrite:

$$\sum_{i=1}^{m} u(i) (d_\pi(i) - d_{\pi'}(i)) + \sum_{i=m+1}^{|\mathcal{T}|} u(i) (d_\pi(i) - d_{\pi'}(i))$$

$$\geqslant u(m) \sum_{i=1}^{m} \left( d_\pi(i) - d_{\pi'}(i) \right) + u(m+1) \sum_{i=m+1}^{|\mathcal{T}|} \left( d_\pi(i) - d_{\pi'}(i) \right), \tag{25}$$

where Equation (25) is obtained by applying the following reasoning. On the one hand, under Equation (22) and under the compatibility of $u$, it holds that $u(1)\left(d_\pi(1) - d_{\pi'}(1)\right) \geqslant u(2)\left(d_\pi(2) - d_{\pi'}(2)\right)$, and by applying a chain reasoning, we can demonstrate that:

$$\sum_{i=1}^{m} u(i)\left( d_\pi(i) - d_{\pi'}(i) \right) \geqslant u(m) \sum_{i=1}^{m} \left( d_\pi(i) - d_{\pi'}(i) \right).$$

On the other hand, under Equation (23) and under the compatibility of $u$, it holds that $u(|\mathcal{T}|)\left(d_\pi(|\mathcal{T}|) - d_{\pi'}(|\mathcal{T}|)\right) \leqslant u(|\mathcal{T}|-1)\left(d_\pi(|\mathcal{T}|-1) - d_{\pi'}(|\mathcal{T}|-1)\right)$, and by applying a similar chain reasoning as before, but in the opposite direction, we get that:

$$\sum_{i=m+1}^{|\mathcal{T}|} u(i)\left( d_\pi(i) - d_{\pi'}(i) \right) \geqslant u(m+1) \sum_{i=m+1}^{|\mathcal{T}|} \left( d_\pi(i) - d_{\pi'}(i) \right).$$

Finally, by applying Equation (24) to Equation (25) we get that:

$$\sum_{i=1}^{|\mathcal{T}|} u(i)\left( d_\pi(i) - d_{\pi'}(i) \right) \geqslant \left( u(m) - u(m+1) \right) \sum_{i=1}^{m} \left( d_\pi(i) - d_{\pi'}(i) \right) \geqslant 0,$$

where the last inequality holds under the compatibility of $u$, thus demonstrating the implication and concluding the proof.

$\square$

**Theorem 5.2.** *Let $\leq_\mathcal{T}$ be a partial order over $\mathcal{T}$ and let $\pi, \pi' \in \Pi$ be two policies. $\pi \leq_\mathcal{T}$-weakly dominates $\pi'$ if and only if, for every realizer $\{\leqslant_{\mathcal{T},i}\}_{i \in [\![m]\!]}$ with $m \in \mathbb{N}$ of $\leq_\mathcal{T}$, it holds that:*

$$\forall i \in [\![m]\!]: \quad \pi' \leqslant_{\Pi,i} \pi,$$

*where $\pi' \leqslant_{\Pi,i} \pi$ (resp. $\pi' <_{\Pi,i} \pi$) denotes that $\pi$ weakly (resp. strictly) $\leqslant_{\mathcal{T},i}$-dominates $\pi'$ (Definition 5.1) w.r.t. the $i$-th total order in the realizer of $\leq_\mathcal{T}$. Furthermore, $\pi \leq_\mathcal{T}$-strictly dominates $\pi'$ if and only if, in addition to the above, it holds that:*

$$\exists j \in [\![m]\!]: \quad \pi' <_{\Pi,j} \pi. \tag{10}$$

*Proof.* We prove the statement for the weak dominance, since the statement for the strict dominance is analogous. We have:

$$\pi' \leq_\Pi \pi \tag{26}$$
$$\Leftrightarrow \forall \boldsymbol{u} \text{ compatible with } \leq_\mathcal{T}: \boldsymbol{J}(\pi; \boldsymbol{u}) - \boldsymbol{J}(\pi', \boldsymbol{u}) \geq \boldsymbol{0} \tag{27}$$
$$\Leftrightarrow \forall \{\leqslant_{\mathcal{T},i}\}_{i \in [\![m]\!]} \text{ realizer of } \leq_\mathcal{T} \forall i \in [\![m]\!] \forall u_i \text{ compatible with } \leq_{\mathcal{T},i}: J(\pi; u_i) - J(\pi', u_i) \geqslant 0 \tag{28}$$
$$\Leftrightarrow \forall \{\leqslant_{\mathcal{T},i}\}_{i \in [\![m]\!]} \text{ realizer of } \leq_\mathcal{T} \forall i \in [\![m]\!]: \pi' \leqslant_{\Pi,i} \pi, \tag{29}$$

where line (27) follows from Definition 4.2, line (28) follows from the fact that a multi-dimensional utility $\boldsymbol{u}$ determines a unique realizer of $\leq_\mathcal{T}$ and from the component-wise order definition, and line (29) is obtained from Definition 4.2. $\square$

**Theorem 5.3.** *Let $\leq_\mathcal{T}$ be a partial order over $\mathcal{T}$ and let $\pi, \pi' \in \Pi$ be two policies. $\pi \leq_\mathcal{T}$-weakly dominates $\pi'$ if and only if, for every linear extension $\leqslant_\mathcal{T}$ of $\leq_\mathcal{T}$, it holds that:*

$$\forall n \in [\![|\mathcal{T}|]\!]: \sum_{i=1}^{n} \left( d_\pi(\psi_{\leqslant_\mathcal{T}}(i)) - d_{\pi'}(\psi_{\leqslant_\mathcal{T}}(i)) \right) \geqslant 0. \tag{11}$$

*$\pi \leq_\mathcal{T}$-strictly dominates $\pi'$ if and only if, in addition to the above, there exists a linear extension $\leqslant'_\mathcal{T}$ of $\leq_\mathcal{T}$ such that:*

$$\exists n \in [\![|\mathcal{T}|]\!]: \sum_{i=1}^{n} \left( d_\pi(\psi_{\leqslant'_\mathcal{T}}(i)) - d_{\pi'}(\psi_{\leqslant'_\mathcal{T}}(i)) \right) > 0. \tag{12}$$

*Proof.* We prove the statement for the weak dominance, as for the strict dominance analogous derivation holds. Recall that the set of all linear extensions of $\leq_{\mathcal{T}}$ is a realizer of $\leq_{\mathcal{T}}$ and that the union of all the realizes of $\leq_{\mathcal{T}}$ is such a set. We have:

$$\pi' \preceq_\Pi \pi \Leftrightarrow \forall \{\leqslant_{\mathcal{T},i}\}_{i \in [\![m]\!]} \text{ realizer of } \leq_{\mathcal{T}} \forall i \in [\![m]\!] : \pi' \leqslant_{\Pi,i} \pi \tag{30}$$

$$\Leftrightarrow \forall \leqslant_{\mathcal{T}} \text{ linear extension of } \leq_{\mathcal{T}} : \pi' \leqslant_\Pi \pi \tag{31}$$

$$\Leftrightarrow \forall \leqslant_{\mathcal{T}} \text{ linear extension of } \leq_{\mathcal{T}} \forall n \in [\![|\mathcal{T}|]\!] : \sum_{i=1}^{n} (d_\pi(i) - d_{\pi'}(i)) \geqslant 0, \tag{32}$$

where Equation (30) follows from Theorem 5.2, Equation (32) follows from Theorem 5.1. $\square$

**Theorem 6.1.** *Let $u, \widehat{u} : \mathcal{T} \to \mathbb{R}^m$ be two $m$-dimensional utilities functions such that $\|u - \widehat{u}\|_F^2 \leqslant \eta^*$. Then, it holds that $\mathcal{L}(u, \widehat{u}) \leqslant 2\sqrt{m\eta^*}$.*

*Proof.* Let $\pi, \widehat{\pi} \in \Pi$ be two Pareto optimal policies w.r.t. $u$ and $\widehat{u}$, respectively. Let $d_\pi$ and $d_{\widehat{\pi}}$ be the corresponding trajectory distributions. We consider matrices $u$ and $\widehat{u}$ both in $\mathbb{R}^{|\mathcal{T}| \times m}$ as constituted by a set of $m$ vectors $(u_j)_{j \in [\![m]\!]}$ and $(\widehat{u}_j)_{j \in [\![m]\!]}$, respectively. Then, for every component $j \in [\![m]\!]$, it holds:

$$J(\pi, u_j) - J(\widehat{\pi}, u_j) = \langle u_j, d_\pi - d_{\widehat{\pi}} \rangle = \langle u_j, d_\pi - d_{\widehat{\pi}} \rangle \pm \langle \widehat{u}_j, d_\pi \rangle \pm \langle \widehat{u}_j, d_{\widehat{\pi}} \rangle$$

$$= \underbrace{\langle u_j - \widehat{u}_j, d_\pi \rangle}_{(A)} + \underbrace{\langle \widehat{u}_j - u_j, d_{\widehat{\pi}} \rangle}_{(B)} + \langle \widehat{u}_j, d_\pi - d_{\widehat{\pi}} \rangle$$

$$\leqslant \underbrace{2\|\widehat{u}_i - u_j\|_\infty}_{(A)+(B)} + \langle \widehat{u}_j, d_\pi - d_{\widehat{\pi}} \rangle,$$

where the inequality follows from the fact that both terms $(A)$ and $(B)$ can be bounded using Holder's inequality with $\|\cdot\|_\infty$ and $\|\cdot\|_1$ and observing that $\|d_\pi\|_1 = \|d_{\widehat{\pi}}\|_1 = 1$. Now, we apply the infimum:

$$\inf_{\widehat{\pi} \in \Pi^*(\widehat{u})} \sum_{j \in [\![m]\!]} (\langle u_j, d_\pi - d_{\widehat{\pi}} \rangle)^+ \leqslant \inf_{\widehat{\pi} \in \Pi^*(\widehat{u})} \sum_{j \in [\![m]\!]} (2\|\widehat{u}_j - u_j\|_\infty + \langle \widehat{u}_j, d_\pi - d_{\widehat{\pi}} \rangle)^+$$

$$\leqslant 2\sqrt{m}\|u - \widehat{u}\|_F + \inf_{\widehat{\pi} \in \Pi^*(\widehat{u})} \sum_{j \in [\![m]\!]} (\langle \widehat{u}_j, d_\pi - d_{\widehat{\pi}} \rangle)^+ \tag{33}$$

$$\leqslant 2\sqrt{m}\|u - \widehat{u}\|_F, \tag{34}$$

where line (33) follows from the application of Cauchy-Schwarz's inequality after having observed that such $\|\cdot\|_\infty$ terms do not depend on $\widehat{\pi}$, and line (34) is due to the fact that the removed term is non-positive by definition of $\widehat{\pi}$ which is Pareto optimal w.r.t. $\widehat{u}$. Replicating the derivation by reversing the roles of $u$ and $\widehat{u}$ leads to the result. $\square$

