# OpenReview forum: "Towards Theoretical Understanding of Sequential Decision Making with Preference Feedback"
_ICML.cc/2025/Conference — ICML 2025 poster_

### Official Review · Reviewer_dYAR · 2025-03-04

**Overall Recommendation:** 2

**Summary:**

This paper considers sequential decision making with preference feedback. The authors build a theoretical formulation linking preferences, utilities (i.e., non-Markovian rewards), and Markovian rewards, and then study the connections between them. First, the authors model preference feedback using a partial (pre)order over trajectories, which enables the presence of incomparabilities. Second, the authors study how a preference relation can be approximated by a multi-objective utility. They introduce a notion of preference-utility compatibility and analyze the computational complexity of this transformation, showing that constructing the minimum-dimensional utility is NP-hard. Third, the authors propose a new concept of preference-based policy dominance that does not rely on utilities or rewards, and analyze the computational complexity of assessing it. Fourth, the authors develop a computationally efficient algorithm to approximate a utility using (Markovian) rewards, and quantify the error in terms of the suboptimality of the optimal policy induced by the approximating reward. This paper aims to lay a foundation for sequential decision making from preference feedback, with promising potential applications in RL from human feedback.

**Claims And Evidence:**

The claims made in this paper are supported by clear and convincing evidence.

**Essential References Not Discussed:**

None

**Experimental Designs Or Analyses:**

There is no experiment in this paper.

**Methods And Evaluation Criteria:**

The proposed methods and evaluation criteria make sense.

**Other Comments Or Suggestions:**

Please see the weaknesses above.

**Other Strengths And Weaknesses:**

Strengths:

1. This paper proposes a theoretical formulation which links preferences, utilities (i.e., non-Markovian rewards), and Markovian rewards, and then study the connections between them.
2. The authors model preference feedback using a partial (pre)order over trajectories and propose a new notion of preference-based policy dominance. In addition, the authors study the computational complexity of the transformation and assessment for the proposed notions in this preference-based and utility-based MDP formulation.

Weaknesses:

1. The writing and readability of this paper should be improved. This paper is hard to follow. The abstract is a bit long.
2. This paper seems to be a pure theoretical work, which defines a MDP based on partial order and proves some of its properties. What is the motivation of the proposed MDP formulation based on partial order and utility? How can this MDP formulation relate and contribute to real-world applications, e.g., the RL with human feedback and LLM applications?
3. There is no new algorithm proposed, and there is no experiment. The contribution is purely theoretical, i.e., a new MDP formulation defined based on partial order and utility, which is limited.

**Questions For Authors:**

Please see the weaknesses above.

**Relation To Broader Scientific Literature:**

This paper is relevant to the literature.

**Theoretical Claims:**

The theoretical results look reasonable, but I didn’t go through every proof.

---

> ### Author Rebuttal · Authors · 2025-03-31
>
> We thank the Reviewer for the time spent reviewing our paper. Below, our answers to the Reviewer's comments and concerns.
>
> > The writing and readability of this paper should be improved. This paper is hard to follow. The abstract is a bit long.
>
> We thank the Reviewer for raising this point. We will make our best efforts to improve the readability of the paper, in particular for Sections 2 and 5, also leveraging the additional available page. We commit to improve it by lightening the notation (moving the not fundamental one to the appendix) and rewriting some parts which are less fluid.
>
> > This paper seems to be a pure theoretical work, which defines a MDP based on partial order and proves some of its properties. What is the motivation of the proposed MDP formulation based on partial order and utility? How can this MDP formulation relate and contribute to real-world applications, e.g., the RL with human feedback and LLM applications?
>
> > There is no new algorithm proposed, and there is no experiment. The contribution is purely theoretical, i.e., a new MDP formulation defined based on partial order and utility, which is limited.
>
> We thank the Reviewer for raising the point. We believe that our theoretical formulation of preference-based MDP (especially with partial orders) is **strongly motivated by real-world applications**. We provide arguments for this statement below:
>
> 1. In the real-world, a **preference feedback is a much more realistic feedback than both rewards (used in RL) and demonstrations (used in Inverse RL)**. Consider, for example, LLM applications: it is quite natural of a human to state which of two proposed answers they prefer, rather than trying to define a reward function for answer generation, or asking the human to demonstrate which answer they would want to receive.
>
> 2. Furthermore, it is as realistic to consider that **a human might not be able to state a clear preference between any pair of proposed trajectories**. Regarding LLM applications, if we were to ask a human to state their preference between two answers, the human may  evaluate different aspects, e.g., the length, the clarity, the correctness, and the harmfulness of the answers. Thus, it is possible that the human may not be able to state a clear preference. This requires modeling **incomparabilities** among trajectories. One natural approach to address this need is to consider **partial order relations** among trajectories. Another example is the well-known problem of autonomous driving, where we want to balance terms of travel time and travel comfort. Clearly, these two objectives are in contrast, as reckless driving brings the passenger to the destination quicker but at the cost of their comfort, whereas a completely comfortable drive may take too much time to reach the destination. By considering a partial order relation over trajectories, we are able to capture the multi-dimensionality of such a problem. The preference-based MDP (PbMDP) framework we propose in this paper allows us to formally define these problems.
>
> 3. Finally, having motivated **why** we need a framework to generalize preferences to allows for incomparabilities, we ask **what** we are able to learn. Thus, we propose novel concepts of **dominance between policies**, and study the computational complexity of evaluating them. The NP-completeness results we propose constitute a computational barrier on **what** is possible to ask of a (practical) algorithm. Then, we study a way to approximate this problem, rendering it computationally tractable, at the cost of an approximation error.
>
> We will leverage the additional page to include a discussion on this point, as we recognize the importance of motivating a novel framework from real-world applications.

---

### Official Review · Reviewer_xt7J · 2025-03-13

**Overall Recommendation:** 3

**Summary:**

This paper establishes a rigorous theoretical framework for sequential decision-making with preference feedback, where agents learn from comparative evaluations of trajectories rather than explicit reward signals. The authors make several key contributions:

1. They model preference feedback using partial preorders between trajectories, enabling the formal characterization of incomparability phenomena that occur when trajectories cannot be meaningfully ranked.

2. The research investigates systematic approaches to approximate preference relations using multi-objective utility functions.

3. The authors develop a novel concept of preference-based policy dominance that operates independently of utility functions or rewards.

4. They present an algorithm that efficiently approximates utilities using Markovian rewards, complete with quantifiable error bounds.

Together, these contributions create a principled framework connecting preferences, utilities, and Markovian rewards in sequential decision-making environments.

**Claims And Evidence:**

Yes.

**Essential References Not Discussed:**

No.

**Experimental Designs Or Analyses:**

This paper contains no experiments.

**Methods And Evaluation Criteria:**

This is a pure theory paper.

**Other Comments Or Suggestions:**

na

**Other Strengths And Weaknesses:**

**Strengths**
- The paper establishes a comprehensive framework that connects preferences, utilities, and rewards in sequential decision-making, filling an important gap in the theoretical understanding of preference-based learning.
- Unlike many existing approaches, the authors explicitly model preferences as partial preorders, allowing for incomparabilities that commonly occur with human preferences but are often overlooked in the literature.
- The work provides valuable insights into the computational complexity of various transformations between preferences, utilities, and rewards, highlighting fundamental challenges in this domain.


**Weaknesses**
- Some theoretical results largely follow or implied by existing known results (e.g., Theorem 4.2). While this work offers new insights for decision making with preference feedback, highlighting the technical novelty of the proof would further strengthen the paper.
- While establishing a solid theoretical foundation, the paper places less emphasis on developing practical algorithms that could be immediately applied to real-world problems.
- The paper is purely theoretical, with no empirical evaluation to validate how well the proposed methods perform in practice compared to existing RLHF methods. This limitation is significant for such an application-driven area.

---

Overall, I appreciate the new theoretical framework developed by the authors and would advocate for acceptance.

**Questions For Authors:**

na

**Relation To Broader Scientific Literature:**

These results build upon previous work in theoretical computer science/game theory and provide new findings for RLHF.

**Theoretical Claims:**

I have reviewed the high-level proof idea and did not find obvious issues.

---

> ### Author Rebuttal · Authors · 2025-03-31
>
> We thank the Reviewer for the time spent reviewing our work and we appreciate the Reviewer's understanding of the relevance of the proposed framework. Below, our answers to the Reviewer's comments.
>
> > Some theoretical results largely follow or implied by existing known results (e.g., Theorem 4.2). While this work offers new insights for decision making with preference feedback, highlighting the technical novelty of the proof would further strengthen the paper.
>
> While we recognize that some of the results, especially those presented in Section 4, can be obtained with non-complex arguments starting from existing ones, **they have never been presented in the literature, as far as we know**. Concerning the **technical novelty**, we report below two examples of results we present that bring technical novelty:
> - Theorem 5.4: its proof involves a reduction to a non-standard NP-complete problem of **topological ordering in weighted DAG** (Gerbner et al., 2016) which requires a non-trivial construction.
> - Theorem 6.1: it **generalizes the bisimulation lemmas** from the case of a scalar utility (or reward) to the case of multi-dimensional utilities. This requires defining a proper index to evaluate suboptimality on the Pareto frontier, which is the function $\mathcal{L}(\boldsymbol{u},\boldsymbol{\widehat{u}})$.
>
> We have rewritten the Original Contribution paragraph to better highlight these aspects.
>
> > While establishing a solid theoretical foundation, the paper places less emphasis on developing practical algorithms that could be immediately applied to real-world problems.
>
> > The paper is purely theoretical, with no empirical evaluation to validate how well the proposed methods perform in practice compared to existing RLHF methods. This limitation is significant for such an application-driven area.
>
> We take the liberty to answer both questions together since our choices are justified by a specific aim. Preference-based RL and/or RLHF is, as the Reviewer notes, "such an application-driven area". However, the understanding of the intimate properties of the preference relations is currently missing. This paper precisely aims to **make a step forward the understanding of the theoretical properties of preference relations** and its limitations (that are significant, as we show) from a computational perspective. This, in our view, represents a fundamental step to, for instance, avoid that practitioners attempt to address problems which turn out to be provably hard (e.g., checking policy dominance for partial orders). This is the reason why we decided not to (1) propose algorithms and (2) conduct an experimental validation. Clearly, building on our work, and aware of the computational limitations, future works should formalize real-world problems according to this setting, and design algorithms capable of solving them, under **certain assumptions**, to achieve convenient computational (and, subsequently, statistical) guarantees.

---

> > ### Comment · Reviewer_xt7J · 2025-04-03
> >
> > Thank you for your response. I believe these clarifications will improve the paper. I will maintain my positive score.

---

### Official Review · Reviewer_6i49 · 2025-03-15

**Overall Recommendation:** 3

**Summary:**

The authors consider the setting of sequential decision-making problems in which only preferences over trajectories are provided, specifically partial (pre)orders. This allows for situations where comparisons of pairs of trajectories are not available (incomparabilities). After several definitions to precisely capture this setting, including the notion of utility-preference compatibility, the authors show how a multiobjective utility function can be constructed from the preorder and that constructing the utility function with the smallest dimensionality is NP-hard. Finally, the authors propose a quadratic program to approximate such a utility function with a reward function for an approximate MDP and provide a bound on the incurred error.

**Claims And Evidence:**

I followed the proofs and could not find any mistakes but this is not my area of expertise.

**Essential References Not Discussed:**

As for inverse RL, I was surprised that the authors did not cite work that directly connects preference elicitation and inverse reinforcement learning.

**Experimental Designs Or Analyses:**

No experiments were conducted.

**Methods And Evaluation Criteria:**

There are no evaluations involving examples or simulations.

**Other Comments Or Suggestions:**

“in the realizer ofthe"

**Other Strengths And Weaknesses:**

This is a theoretical paper that makes notions of utility-preference compatibility, policy dominance, and the connection between partial preorders on trajectories, utility functions, and reward functions in an approximate MDP precise, shows that establishing policy dominance in the very general case considered here is NP-hard, and provides a constructive algorithm for reward functions in an approximate MDP. This relates to preference-based RL and RL from human feedback.

It is difficult for me to assess how much of an advancement in the field this is, as I am not an expert in this area.

**Questions For Authors:**

Are there any problems and datasets that could be used to construct Markovian reward functions solving the quadratic program (eq. 13) and thereby provide a sense of the applicability of the present algorithm and the value of the error bound?

**Relation To Broader Scientific Literature:**

This is not my area of expertise, and I am not knowledgeable enough about preference-based RL and RL from human feedback. I was wondering about any potential canonical problems and datasets that could be used to construct Markovian reward functions solving the quadratic program (eq. 13) and thereby providing a sense of the applicability of the present algorithm and the value of the provided error bound.

**Theoretical Claims:**

I followed the proofs and could not find any mistakes but this is not my area of expertise.

---

> ### Author Rebuttal · Authors · 2025-03-31
>
> We thank the Reviewer for the time spent reviewing our work, for understanding the relevance of the QP and of the error bound. Below, our answers to the Reviewer's questions.
>
> > As for inverse RL, I was surprised that the authors did not cite work that directly connects preference elicitation and inverse reinforcement learning.
>
> We thank the Reviewer for raising this point. We have not included methods specific to the Inverse RL field in the related works of this paper due to the following reasoning:
>
> 1. **IRL focuses on observing a behavior** that is assumed to be optimal and learning a reward function, whereas, according to our framework, we consider the case in which the **agent interacts with the environment, asking for preference feedback** to be used to estimate a reward and to improve its policy. For this reason, the two frameworks are different;
> 2. Preference elicitation and IRL are related to the learning aspect of the problem, and, thus, to the statistical complexity of doing so, whereas, in this work, we focus on the different concepts of optimality and the computational complexity of estimating multi-dimensional utility and reward functions. In other words, we focus more on **computational complexity** rather than on **statistical complexity**. As we briefly discuss in the future works, one possible direction is to address learning with preference feedback when incomparabilities are possible. When tackling such a direction, it will then be necessary to compare with the literature of preference elicitation and IRL.
>
> However, we acknowledge that there exist works that combine both preference elicitation and IRL, such as (Rothkopf and Dimitrakakis, 2011). We added a discussion on this in the related works.
>
> Rothkopf, C. A., and Dimitrakakis, C. Preference elicitation and inverse reinforcement learning. ECML-PKDD 2011.
>
> > Are there any problems and datasets that could be used to construct Markovian reward functions solving the quadratic program (eq. 13) and thereby provide a sense of the applicability of the present algorithm and the value of the error bound?
>
> In principle, existing RL benchmarks could be adapted to accomodate the problem of solving the QP to estimate an approximated reward function. This can be done either by querying preference from real humans (see, e.g., Christiano et al., 2017) or by defining a synthetic expert (see, e.g., Akrour et al., 2012). One example of existing datasets which can be adapted to allow for incomparabilities are those based on the OpenAI Gym, which can be found in Minari (https://minari.farama.org/main/), from the Farama Foundation.
> Regarding LLMs, there exist datasets which, with the necessary adaptations, could be considered for what the Reviewer suggests. One such example is the *Preference Dissection* dataset (Li et al., 2024), which contains questions, pairs of answers, and features for each answer. Although the reported human preferences do not admit incomparabilities, synthetic preferences could be generated considering the answers and their features. An additional example is the *PKU-SafeRLHF* dataset (Ji et al., 2024), which contains questions and answers labelled in terms of harmfulness and correctness. Again, it would be necessary to generate synthetic preferences.
> In conclusion, we believe that the definition of a standardized benchmark containing both standard RL tasks and language tasks, together with human-labelled datasets and synthetic parametric experts would be beneficial for the evaluation of future approaches to the problem of learning from preferences with incomparabilities.
>
> Christiano, P. F., Leike, J., Brown, T., Martic, M., Legg, S., and Amodei, D. Deep reinforcement learning from human preferences. NeurIPS 2017.
>
> Akrour, R., Schoenauer M., and Sebag M. April: Active preference learning-based reinforcement learning. ECML-PKDD 2012.
>
> Li, J., Zhou, F., Sun, S., Zhang, Y., Zhao, H., and Liu, P. Dissecting human and llm preferences. arXiv preprint arXiv:2402.11296, 2024.
>
> Ji, J., Hong, D., Zhang, B., Chen, B., Dai, J., Zheng, B., Qiu, T., Li, B., and Yang, Y. PKU-SafeRLHF: Towards Multi-Level Safety Alignment for LLMs with Human Preference. arXiv preprint arXiv:2406.15513, 2024.

---

### Official Review · Reviewer_g3qH · 2025-03-18

**Overall Recommendation:** 3

**Summary:**

This paper aims to build a theoretical basis linking the preference-based MDP, the utility-based MDP, and the reward-based MDP. Specifically, this paper formulates these three settings in Section 3, and discusses the connections between the preference-based MDP and the utility-based MDP in Section 4. In Section 5, this paper discusses the dominance and optimality with preferences; and finally, in Section 6, it discusses the relationship between the utility-based MDP and the reward-based MDP.

**Claims And Evidence:**

To the best of my knowledge, the claims made in the submission are supported by clear and convincing evidence.

**Essential References Not Discussed:**

To the best of my knowledge, No.

**Experimental Designs Or Analyses:**

Not applicable.

**Methods And Evaluation Criteria:**

This paper does not have any experiment results.

To the best of my knowledge, the theoretical claims (theorems) in this paper seem to be correct.

**Other Comments Or Suggestions:**

This paper has done a good job of literature review.

**Other Strengths And Weaknesses:**

I have some concerns about the novelty and significance of this paper. Specifically,

1) It seems that some parts of this paper are well-known results from the classical choice theory and existing work, such as Theorem 4.2. Might the authors clearly explain in the rebuttal what are the new results of this paper and what are the existing results?

2) My understanding is that this paper has discussed many different issues related to preference-based MDPS, utility-based MDPs, reward-based MDPs, as well as their connections. Several theorems have been developed; however, it seems that none of them is very hard to prove, and none of them is really counter-intuitive. This might reduce the significance of this paper.

In addition, from the perspective of writing, this paper is mathematically too heavy, and is not easy to read.

**Questions For Authors:**

Please try to address the questions/weaknesses listed above.

-----------------------------------------------

I have read the authors' rebuttal, which has addressed some of the concerns listed above, especially the concerns about the novelties and technical contributions of this paper. I will increase my recommendation from 2 to 3.

**Relation To Broader Scientific Literature:**

This paper has done a good job of literature review. However, I have some concerns about the novelty and significance of this paper, which I will list below.

**Theoretical Claims:**

I have checked the proofs in a high level. To the best of my knowledge, the proofs seem to be correct.

---

> ### Author Rebuttal · Authors · 2025-03-31
>
> We thank the Reviewer for the time spent reviewing our work. Below, our answers to the Reviewer's questions and concerns.
>
> > It seems that some parts of this paper are well-known results from the classical choice theory and existing work, such as Theorem 4.2. Might the authors clearly explain in the rebuttal what are the new results of this paper and what are the existing results?
>
> To the best of our knowledge, **all the results presented in the paper are novel**. While we recognize that some of them, especially those presented in Section 4, can be obtained with non-complex arguments starting from existing ones, they have never been presented in the literature, as far as we know. Specifically, considering Theorem 4.2, while it is known that **computing the order-dimension is NP-hard** (Yannakakis, 1982), we claim that also **computing a minimal compatible utility is NP-hard**. This is proved with a simple, we recognize, but still novel, **reduction** to the problem of computing the order-dimension. We kindly ask the reviewer to provide references that already report any of the results we provide in this paper.
>
> We refer the Reviewer also to the itemize in the answer below for further discussion.
>
> > My understanding is that this paper has discussed many different issues related to preference-based MDPS, utility-based MDPs, reward-based MDPs, as well as their connections. Several theorems have been developed; however, it seems that none of them is very hard to prove, and none of them is really counter-intuitive. This might reduce the significance of this paper.
>
> We  honestly **disagree with the Reviewer on the fact that if a theorem is "not very hard to prove" or "really counter-intuitive" then this reduces its significance**. Beside being quite subjective notions, we wonder how many papers accepted at top-conferences (like ICML) really contain theorems that are "very hard to prove" and "really counter-intuitive". Just think to the bandit literature where the regret bound is often known before proving it (not counter-intuitive) and the proofs follow minor variations of techniques established by 20 years (not very hard to prove). Nevertheless, we report below two examples of theorems, in our opinion, that are either counter-intuitive or hard to prove:
> - Theorem 5.1: is *counter-intuitive* since it is not obvious that the condition of Definition 5.1, which involves an **existential quantification** over the compatible utilities (which are an infinite continuous set), can be verified by checking a finite number of inequalities.
> - Theorem 5.4: is *not simple to prove* since it involves a reduction to a non-standard NP-complete problem of **topological ordering in weighted DAG** (Gerbner et al., 2016) which requires a non-trivial construction. Furthermore, in the authors' opinion, the fact that checking policy dominance in a partial order is NP-complete is quite *counter-intuitive*.
>
> Moreover, there are results that are significant beyond this work. For example, Theorem 6.1 generalizes the **bisimulation lemmas** from the case of a scalar utility (or reward) to the case of multi-dimensional utilities. This requires defining a proper index to evaluate suboptimality on the Pareto frontier (function $\mathcal{L}(\boldsymbol{u},\boldsymbol{\widehat{u}})$).
>
> > In addition, from the perspective of writing, this paper is mathematically too heavy, and is not easy to read.
>
> We thank the Reviewer for raising the point. We will make our best efforts to improve the redability of the paper, in particular for Sections 2 and 5, also leveraging the additional available page. We commit to improve it by lightening the notation (moving the not fundamental one to the appendix) and rewriting some parts which are less fluid.

---

> > ### Comment · Reviewer_g3qH · 2025-04-09
> >
> > Thanks for the rebuttal and clarifications. The rebuttal has partially addressed my concerns, especially my concerns on the novelties and technical contributions of this paper. I will increase my overall recommendation to 3.

---

### Decision · Program_Chairs · 2025-05-01

**Decision:**

Accept (poster)

**Comment:**

This paper provides a theoretical framing and analysis of problems that connect trajectory-based preferences, trajectory-based utilities, and Markov rewards. The work is largely theoretical, introducing several new notions and proving several complexity and approximation results. The overall contribution is framed as providing a foundation for sequential decision making from preference feedback. Overall, the reviewers appreciated the potential contributions of the paper, but raised several questions concerning the overall significance and surprise of the results, the lack of empirical evaluation (some of the results do avail themselves of such), and the fact that the paper, while using RLHF as one of its motivations, does not make anything other than superficial connections to that literature. The authors response (and other comments) were helpful, but did not fully alleviate the concerns of significance. Overall, this paper makes a reasonable contribution, but whether it meets the ICML bar is at issue.